# Influence and Control of SARS-CoV-2 Transmission under Two Different Models

**Xubin Gao** [1,†], **Shuang Chen** [1,*,†], **Qiuhui Pan** [2,3,*,†], **Mingfeng He** [2,3,†] and **Leilei Qu** [4,†]

1   School of Information Science & Engineering, Dalian University, Dalian 116600, China
2   School of Mathematical Science, Dalian University of Technology, Dalian 116024, China
3   School of Innovation and Entrepreneurship, Dalian University of Technology, Dalian 116024, China
4   School of Information Science & Engineering, Dalian Ocean University, Dalian 116024, China
*   Correspondence: chenshuang0707@163.com (S.C.); qhpan@dlut.edu.cn (Q.P.);
    Tel.: +86-0411-84707445-6602 (ext. 116024) (Q.P.)
†   These authors contributed equally to this work.

**Abstract:** SARS-CoV-2 is transmitted by contacting; however, the virus is so active that it can attach to objects and be transmitted from objects to humans via such contacting. The virus, which spreads through some living or inanimate-mediated processes, is more dangerous. On the basis of the routine contact transmission of infectious diseases, this paper further discusses the scope and efficiency of infectious diseases with indirect transmission. Through the study of two different transmission routes, the dynamic models of infectious diseases were constructed. The propagation of these two processes is theoretically studied using a differential equation model and stability analysis theory, and some actual virus propagation processes are simulated by numerical solutions. The prevention and control methods of infectious diseases are given, which lay the theoretical foundation for the discussion of related problems in practical application.

**Keywords:** SARS-CoV-2; inanimate-mediated processes; differential equation model

## 1. Introduction

In March 2020, the World Health Organization declared COVID-19 a global pandemic [1–3], a disease caused by the severe acute respiratory syndrome Coronavirus 2 (SARS-CoV-2) [2]. The novel Coronavirus is considered to be contagious during the incubation period, which is usually observed for three–seven days, but could last for up to 14 days [4,5]. Numerous studies have reported the transmission of SARS-CoV-2 from patients who have not yet developed symptoms, and COVID-19 can have multiple clinical manifestations [4,6]. The most common symptoms of the disease include fever, cough, fatigue, muscle aches, headaches and difficulty breathing [5,7–9].

In the latest COVID-19 outbreak, most patients developed a very mild, self-limiting viral respiratory illness. The statistical results show that the average recovery time of mild and moderate patients was $10.63 \pm 1.93$ days, and for severe patients, it was $18.70 \pm 2.50$ days, that is, two to three weeks [10–12]. The continued outbreak of COVID-19, which began in 2019, has posed huge challenges to people's livelihoods. The existing literature mainly uses mathematical models to predict the incidence level, transmission trend, peak time and the impact of prevention and control measures on the pandemic [13]. These models usually assume that the population is well mixed. As the results of these models are sensitive to initial values and assumptions, there are large differences between the models, particularly in estimating the basic regeneration number (the number of people infected per infected person in a susceptible population), which suggests that 25 to 70 per cent of the population will eventually become infected. The GEP model proposed by Salgotra et al. can be used as a benchmark for time series prediction [14]. As of 14 April 2022, the cumulative number of COVID-19 cases associated with COVID-19 has reached 500,186,525 globally, with more

than 6,190,349 cumulative deaths. It is still in the pandemic phase, with more than 900,000 new cases per day [15]. Model results suggest that only strict lockdown measures and social distancing restrictions can effectively control and contain the pandemic [16–18]. In order to study and control the spread of infectious diseases, people have made unremitting efforts in many aspects. Different degrees of progress have been made regarding the study of the origin [19–21], transmission [13,22–26], control [27–30] and treatment [10,31,32] of the virus.

What are the introduction mechanisms and transmission routes of 2019-nCoV? Most importantly, 2019-nCoV is a novel enveloped single-stranded zoonotic RNA virus belonging to the genus β-coronavirus [9]. It has been reported to share a 96% homology with bat coronavirus, 91% homology with pangolin coronavirus, and 79% homology with SARS coronavirus [33,34]. The Centers for Disease Control and Prevention (CDC) reports that respiratory droplets, aerosols, and close contact (less than 1 m) are the primary modes of transmission [9,35,36]. Droplets can be spread by coughing, sneezing, talking, singing, and touching mucous membranes, especially those of the nose, eyes, or mouth. They stay in the air longer and can travel distances of more than one meter. Aerosol-generating procedures can also lead to airborne transmission [37].

Khan M.A. and Atangana A. [38] devised a model on the assumption that the seafood market has a sufficient number of infection sources with which to infect people. Pei et al. [39] developed and validated an ensemble forecast system for predicting the spatiotemporal spread of influenza that readily uses accessible human mobility data and a metapopulation model. Dayong Zhou et al. [40] presented the concept of safe medical resources, i.e., the minimum amount of medical resources needed to prevent their overburden, and explore the impacts of medical resources on the spread of emerging, self-limiting infectious diseases.

However, the unusual route of transmission in the recent SARS-CoV-2 outbreak has drawn attention. The initial transmission process was only common person-to-person contact transmission, but since SARS-CoV-2 has rapid mutation and transmission speeds, a long incubation period and a strong concealment [41,42], it has caused great problems to human epidemic prevention [43].

Recently, new changes have taken place in the transmission route of SARS-CoV-2. In many areas, seafood, fruit, clothing, express delivery, cold-chain food, etc., have been found to be positive and have transmitted the virus to humans [44,45]. This indicates that SARS-CoV-2 has evolved from the original human-to-human transmission process into an object-to-human transmission process. In other words, SARS-CoV-2 can spread through some kind of inanimate-mediated process. Due to the diversity of mediums, this new route of transmission has caused great problems to the prevention and control of the pandemic. The frequency and range of disinfection have become a problem that must be considered.

In this paper, differential equations are used to establish two models. The first model only discusses the direct viral transmission to humans in the initial stage without detection and prevention and control. The second model introduces mediators (living or inanimate) between living organisms (humans or wild animals) and healthy humans, and indirectly transmits the virus to humans.

In short, this is a process of viral human–object–human transmission. While there was no direct human-to-human transmission, the diversity of inanimate-mediated processes made the transmission more hidden, increasing the difficulty of detection and elimination, and provided enough time for the spread of the virus. The influence of intermediate media on virus transmission in the early stage of transmission was also investigated. This paper discusses the role of inanimate-mediated processes in the transmission chain and provides a theoretical basis of the frequency and range of inanimate-mediated disinfection, so as to provide a method of blocking the virus transmission chain.

## 2. Model

In order to compare the influence of an inanimate-mediated process on virus transmission, two models were established, direct transmission and indirect transmission, and they are described in this section.

### 2.1. Direct Transmission Model

We consider a population of humans within a certain range, and there is an invasive population of a certain disease source, where $W$ represents the pathogenic population; this could be animals or humans. The pathogenic population is limited by the carrying capacity $K$, which propagates with $b_w$ and exits the system with $d_w$. For virus transmission in humans using the SEI model, the subscript $h$ denotes the human population; $S_h$, $E_h$, $I_h$ are susceptible, incubation, and infected population, respectively. The incubation population is not easy to distinguish, but it is infectious. Infected population means diagnosis, and no longer contact with a susceptible population.

The direct transmission process is represented by the following transmission diagram:
According to Figure 1, the ordinary differential equations are established as follows:

$$\begin{cases} \dot{W} = b_w W \left(1 - \frac{W}{K}\right) - d_w W \\ \dot{S_h} = A - \beta_h S_h (E_h + I_h) - \beta_w S_h W - d_h S_h \\ \dot{E_h} = \beta_h S_h (E_h + I_h) + \beta_w S_h W - \alpha E_h - d_h E_h \\ \dot{I_h} = \alpha E_h - (d_h + d) I_h \end{cases}$$

with the initial condition $W(0) = 100$, $S_h(0) = 1,000,000$, $E_h(0) = 0$, $I_h(0) = 0$.

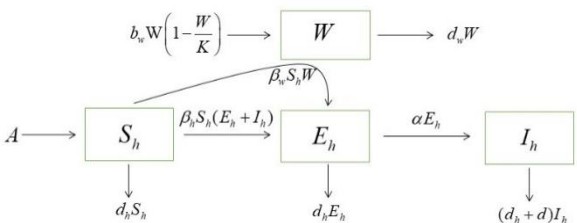

**Figure 1.** Transforming relationships with direct transmission.

### 2.2. Indirect Transmission Model

As above, use $W$ to denote the pathogenic population, subscript $v$ to denote the inanimate-mediated processes, and subscript $h$ to denote humans. The inanimate-mediated processes were divided into susceptible $S_v$ (without the virus) and infected $I_v$ (with the virus). In inanimate-mediated processes, the virus is mainly derived from contact with the pathogenic population, and the transmission rate is $\beta_w$.

Humans are also divided into susceptible ($S_h$), incubation ($E_h$), and infected ($I_h$) population. Considering that humans are not in direct contact with the pathogenic population, they can become infected from exposure to an infected inanimate-mediated processes, and the transmission rate is $\beta$. The remaining parameter settings are shown in Table 1.

The process of person–object–person transmission is represented by the following transmission diagram:

At the same time, the corresponding differential equation model is established, according to Figure 2.

$$
\begin{cases}
\dot{W} = b_w W \left(1 - \frac{W}{K}\right) - d_w W \\
\dot{S}_v = b_v S_v \left(1 - \frac{S_v + I_v}{N}\right) - \beta_v S_v I_v - \beta_w S_v W - d_v S_v \\
\dot{I}_v = \beta_v S_v I_v + \beta_w S_v W - (d_v + d_1) I_v \\
\dot{S}_h = A - \beta S_h I_v - \beta_h S_h (E_h + I_h) - d_h S_h \\
\dot{E}_h = \beta S_h I_v + \beta_h S_h (E_h + I_h) - \alpha E_h - d_h E_h \\
\dot{I}_h = \alpha E_h - (d_h + d) I_h
\end{cases}
$$

with the initial condition $W(0) = 100$, $S_v(0) = 100$, $I_v(0) = 0$, $S_h(0) = 1,000,000$, $E_h(0) = 0$, $I_h(0) = 0$.

**Table 1.** The parameters in models.

| Parameters | Definitions | Value | Unit |
|------------|-------------|-------|------|
| $K$ | Carrying capacity of pathogenic population | 5000 | Quantity |
| $b_w$ | Incidence rate of pathogenic population | 0.01 | Day$^{-1}$ |
| $d_w$ | Removal rate of pathogenic population | 0.01 | Day$^{-1}$ |
| $A$ | Recruitment rate of human | 1000 | Quantity |
| $d_h$ | Removal rate of humans | $3.6 \times 10^{-8}$ | Day$^{-1}$ |
| $\beta_h$ | Transmission rate among human | $3 \times 10^{-8}$ | Day$^{-1}$ |
| $\beta_w$ | Transmission rate from pathogenic population to human | $1 \times 10^{-8}$ | Day$^{-1}$ |
| $\alpha$ | Average latency period | $\frac{1}{14}$ | Dimensionless |
| $d$ | Removal rate of infected human | 0.09817 | Day$^{-1}$ |
| $N$ | Carrying capacity of inanimate-mediated | 5000 | Quantity |
| $\beta_v$ | Transmission rate among inanimate-mediated | $1 \times 10^{-6}$ | Day$^{-1}$ |
| $b_v$ | The growth rate of inanimate-mediated | 0.001 | Day$^{-1}$ |
| $d_v$ | Removal rate of inanimate-mediated | 0.0001 | Day$^{-1}$ |
| $d_1$ | Disinfection rate | 0–1 | Day$^{-1}$ |
| $\beta$ | Transmission rate from inanimate-mediated to human | 0.01 | Day$^{-1}$ |

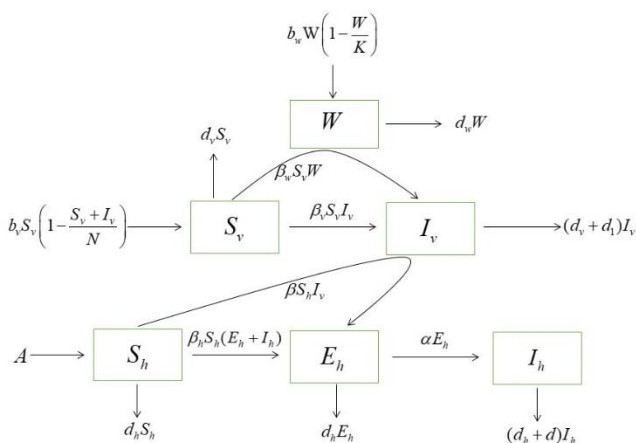

**Figure 2.** Transforming relationships with indirect transmission.

## 3. Results

### 3.1. Equilibrium Stability and Basic Reproduction Number

By using the stability analysis of the equilibrium point of the differential equation model, the long-term stable state of the model can be studied.

For model 1, the following equilibrium points are found:

$$E_1(0, \frac{A}{d_h}, 0, 0)$$

$$E_2(0, \frac{(\alpha + d_h)(d_h + d)}{\beta_h(\alpha + d_h + d)}, \frac{A\beta_h(\alpha + d_h + d) - d_h(\alpha + d_h)(d_h + d)}{\beta_h(\alpha + d_h + d)(\alpha + d_h)}, \frac{\alpha(A\beta_h(\alpha + d_h + d) - d_h(\alpha + d_h)(d_h + d))}{\beta_h(\alpha + d_h + d)(\alpha + d_h)(d_h + d)})$$

$$E_3(W^*, S_h^*, E_h^*, I_h^*)$$

where $W^* = \frac{K(b_w - d_w)}{b_w}$, $S_h^* = S_h^*$, $E_h^* = \frac{A - S_h^* d_h}{\alpha + d_h}$, $I_h^* = \frac{\alpha(b_w(A - S_h^* d_h) - S_h^* \beta_w K(b_w - d_w))}{S_h^* \beta_h b_w(\alpha + d + d_h)}$.

$S_h^*$ is the positive real root of equation:

$$\beta_h b_w d_h(\alpha + d + d_h)S_h^{*2}$$
$$+(-\beta_w K(b_w - d_w)(d + d_h)(\alpha + d_h) - b_w d_h(\alpha + d_h)(d + d_h) - A\beta_h b_w(\alpha + d + d_h))S_h^*$$
$$+Ab_w(d + d_h)(\alpha + d_h) = 0$$

To prove the stability of the disease-free equilibrium, the Jacobian matrix can be calculated:

$$J = \begin{bmatrix} b_w(1 - \frac{W}{K}) - \frac{b_w W}{K} - d_w & 0 & 0 & 0 \\ -\beta_w S_h & -\beta_w W - \beta_h(E_h + I_h) - d_h & -\beta_h S_h & -\beta_h S_h \\ \beta_w S_h & \beta_h(E_h + I_h) + \beta_w W & \beta_h S_h - \alpha - d - d_h & \beta_h S_h \\ 0 & 0 & \alpha & -d - d_h \end{bmatrix}$$

For the disease-free equilibrium $E_1(0, \frac{A}{d_h}, 0, 0)$, eigenvalues of the Jacobian determinant are $b_w - d_w$, $-d_h$, $-\alpha - d - d_h$ and $\frac{A\beta_h - dd_h - d_h^2}{d_h}$.

For the eigenvalue $b_w - d_w$, when $b_w - d_w < 0$, the disease-free equilibrium is asymptotically stable.

In other words, when the incidence rate of the pathogenic population is lower than the removal rate, the disease-free equilibrium is stable. This situation requires control of the population of the pathogenic population.

In other cases, if no action is taken, the pathogenic population will eventually carry the disease. Due to the number of human interventions, we also focus on disease development at the beginning of model development (60 days).

We are mainly concerned with the state of transmission in its early stages. Using the stability theory of differential equations, the equilibrium points of model 2 are discussed.

$$E_1(0, 0, 0, \frac{A}{d_h}, 0, 0)$$

$$E_2(\frac{K(b_w - d_w)}{b_w}, 0, 0, \frac{A}{d_h}, 0, 0)$$

$$E_3(0, \frac{N(b_v - d_v)}{b_v}, 0, \frac{A}{d_h}, 0, 0)$$

$$E_4(0, 0, 0, S_h^1, E_h^1, I_h^1)$$

$$E_5(\frac{K(b_w - d_w)}{b_w}, 0, 0, S_h^1, E_h^1, I_h^1)$$

$$E_6(0, \frac{N(b_v - d_v)}{b_v}, 0, S_h^1, E_h^1, I_h^1)$$

$$E_7(0, \frac{d_v + d_1}{\beta_v}, \frac{\beta_v N b_v - \beta_v N d_v - b_v d_1 - b_v d_v}{\beta_v(\beta_v N + b_v)}, S_h^*, E_h^*, I_h^*)$$

$$E_8(\frac{K(b_w - d_w)}{b_w}, S_v^{**}, I_v^{**}, S_h^{**}, E_h^{**}, I_h^{**})$$

The representation of each symbol is shown in Appendix A.

We use a next-generation matrix to calculate the basic reproduction number of model 1. There is a disease-free equilibrium point $E_1(0, \frac{A}{d_h}, 0, 0)$. From model 1, it can be obtained that:

$$F = \begin{bmatrix} \beta_h S_h & \beta_h S_h & \beta_w S_h \\ 0 & 0 & 0 \\ 0 & 0 & 0 \end{bmatrix}; \quad V = \begin{bmatrix} \alpha + d_h & 0 & 0 \\ -\alpha & d_h + d & 0 \\ 0 & 0 & d_w - b_w + \frac{2b_w W}{K} \end{bmatrix}$$

$$R_0 = \rho(FV^{-1}) = \frac{\beta_h S_h(\alpha + d + d_h)}{(d + d_h)(\alpha + d_h)}$$

As in model 1, we obtained the following for model 2:

$$F = \begin{pmatrix} \beta_h S_h & \beta S_h & \beta_h S_h & 0 \\ 0 & \beta_v S_v & 0 & \beta_w S_v \\ 0 & 0 & 0 & 0 \\ 0 & 0 & 0 & 0 \end{pmatrix}$$

$$V = \begin{pmatrix} \alpha + d_h & 0 & 0 & 0 \\ 0 & d_v + d_1 & 0 & 0 \\ -\alpha & 0 & d_h + d & 0 \\ 0 & 0 & 0 & d_w - b_w + \frac{2W b_w}{K} \end{pmatrix}$$

For the disease-free equilibrium point $E_3(0, \frac{N(b_v - d_v)}{b_v}, 0, \frac{A}{d_h}, 0, 0)$, the basic reproduction number of model 2 is: $R = \max\{R_1, R_2\}$, where $R_1 = \frac{\beta_h S_h(\alpha + d + d_h)}{(\alpha + d_h)(d + d_h)}$, $R_2 = \frac{\beta_v S_v}{d_v + d_1}$.

In the long term, if nothing is added, the model will eventually stabilize the co-existence of the disease. In the actual process, when the spread of the disease was isolated, disinfection, treatment, vaccine and other effective measures were used to balance the condition. We do not pay attention to how this model functions in the stable state; instead, we pay more attention to the incipient stage of the disease.

In the next section, we compare the numerical solutions of the two models, focusing on the transmission status of the disease at the beginning stage, and via a comparison, we investigate whether changes occur in the early stage of infection when SARS-CoV-2 has a new transmission route of person–object–person, and how to control such changes.

### 3.2. Numerical Simulation

In the actual experiment, we are not interested in the stability of the model after an infinite amount of time. Based on prevention and treatment methods and the status of COVID-19 at the present stage, it is vital to understand the disease transmission status in the early stage of virus development.

The most important steps are to reduce the scope, speed and number of people infected at an early stage. In this part of the discussion, we mainly discuss the influence of two different parameters on two different models.

At the same time, we only focus on the change in the population of pathogen and different human populations and do not consider the change in the number of inanimate-mediated populations. The changes in the numbers of various groups during the first two months of infection (60 days) were also considered.

Model 1 is derived from a traditional infectious disease prevention and control model. In order to reduce or even block the impact of infectious sources on human beings, combined with the current epidemic prevention measures, the influence of two parameters, $d_w$ and $d$, on disease prevention and control is given.

If parameter $d_w$ increases, the removal rate of pathogenic population increases, and the mortality rate is certain. This parameter generally refers to increased efforts to prevent the entry of the pathogenic population into the system. In terms of the epidemic prevention

and control situation, it means that a nation has adopted strict border control and effective prevention and control of pathogenic populations that are imported from abroad.

The pathogen population that infects the immune system will be forced into isolation and other measures will force it out of this system.

It is clearly observed in Figure 3 that, with the increase in $d_w$, the increase in the removal rate of the pathogenic population, the number of incubation populations and infected populations are effectively controlled in the early stage of the infection, and isolation measures, which were adopted for this externally imported pathogenic population, are very effective.

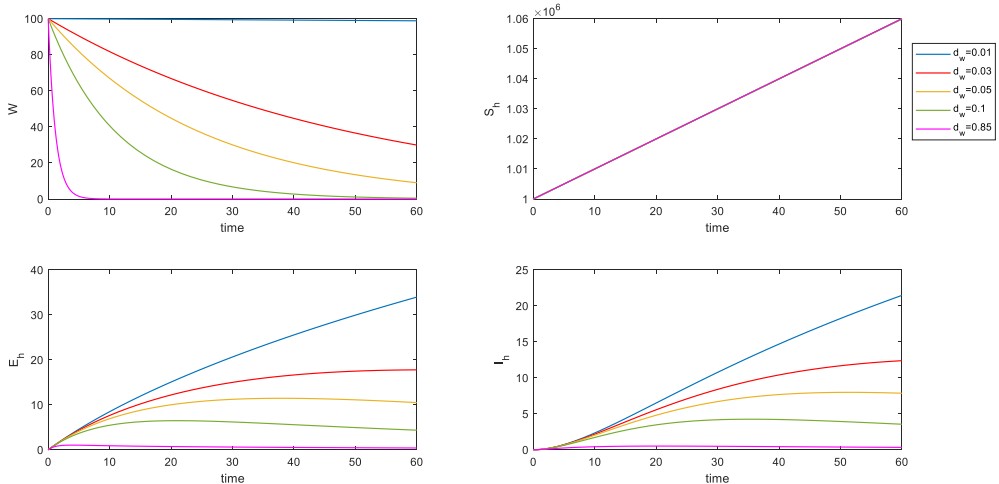

**Figure 3.** Changes in the number of $W$, $S_h$, $E_h$, and $I_h$ with different $d_w$ in model 1.

In Figure 4, the role of parameter $d$ in model 1 can be observed. Parameter $d$ represents the removal of $I_h$. Due to the improvement in treatment methods, the extra mortality rate of patients is low. Therefore, $d$ mainly refers to the proportion of patients who are forced into social isolation, which is determined by the rate of diagnosis.

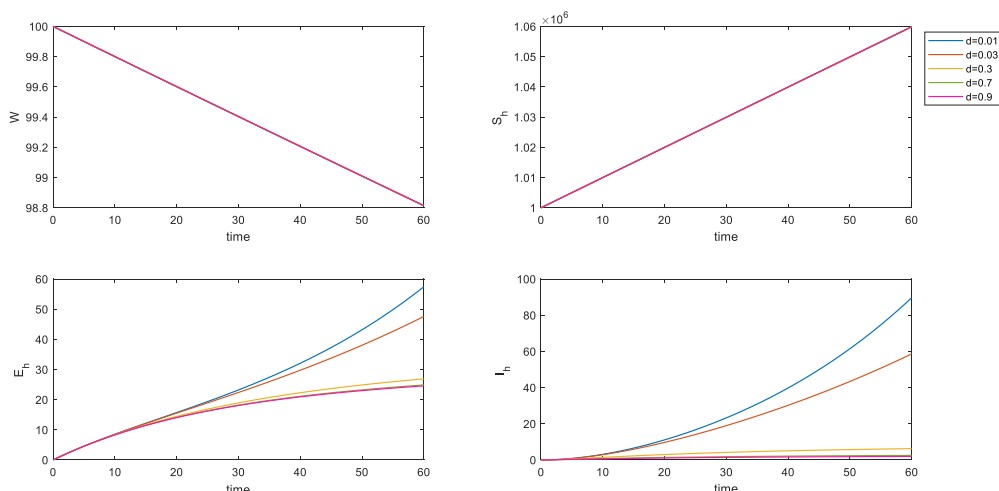

**Figure 4.** Changes in the number of $W$, $S_h$, $E_h$, and $I_h$ with different $d$ in model 1.

Thus, the increase in $d$ mainly indicates an improvement in the diagnosis rate.

The results show that, in the early stage of infection, increasing the rate of confirmed diagnosis was especially effective for the proportion control of infected patients, but for the patients in the early stage, because direct diagnosis isolation was not considered, the control of the number of patients in the incubation period was effective; however, this effect was not as strong as the effect of controlling confirmed patients.

Regarding disease transmission via inanimate-mediated processes, we are still primarily concerned with the initial changes in the number of $S_h$, $E_h$, and $I_h$. We need to understand the role that the addition of mediators plays in disease transmission. Does it influence the effect of the parameters? How much does it affect them? In the following, we use the same parameters, as shown in Model 1, to discuss the influence of the changes in parameters $d_w$ (Figure 5) and $d$ (Figure 6) on Model 2.

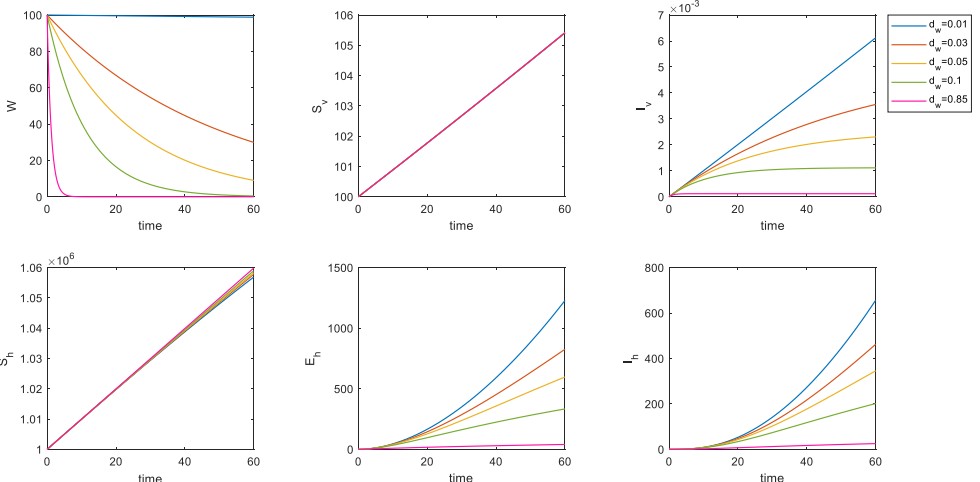

**Figure 5.** Changes in the numbers of $W$, $S_h$, $E_h$, and $I_h$ with different $d_w$ in model 2.

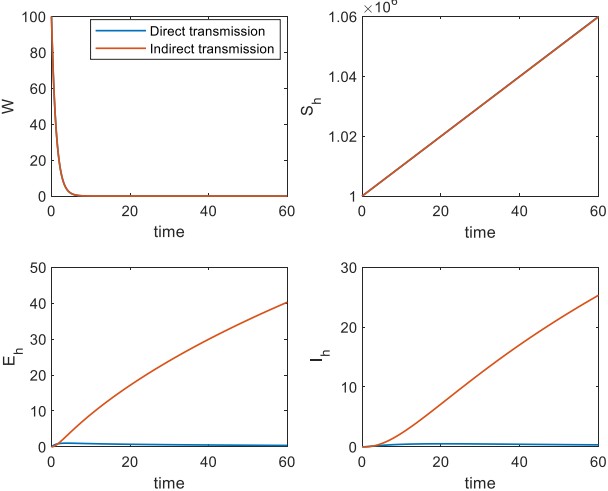

**Figure 6.** Changes in the numbers of $W$, $S_h$, $E_h$, and $I_h$ with $d_w = 0.85$ in model 1 (direct transmission) and 2 (indirect transmission).

The influence of $d_w$ in the two models and the change trend in population are essentially the same. However, through a comparison of the three groups of different $d_w$, it is not clear that, when $d_w$ is the same, regardless of whether it is a direct or indirect infection, there is essentially no difference in the level of pathogenic populations and susceptible populations. However, there is a significant difference between the number of incubation populations and infected populations. Indirect infection is more conducive to the spread of the disease. In the initial stage, even if $d_w = 0.85$, the number of incubation populations and infected populations can still reach about 40 and 20, respectively. When $d_w$ is smaller, the numbers of both populations are larger, and it is essentially impossible to control as a direct infection.

In Figure 7, direct infection and indirect infection have very little difference in the number of pathogenic populations and susceptible populations, but there is a significant difference between the number of infected populations in general and in the incubation period.

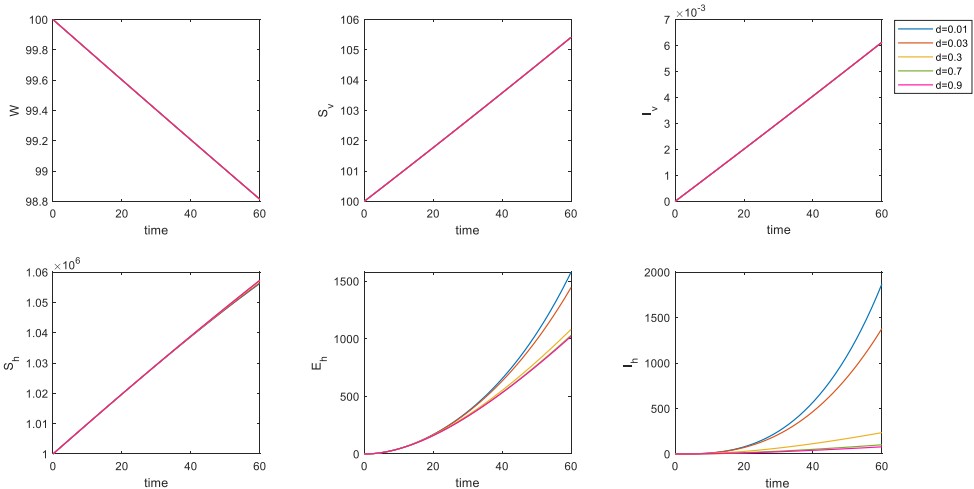

**Figure 7.** Changes in the numbers of $W$, $S_h$, $E_h$, and $I_h$ with different $d$ in model 2.

In Figure 8, for the same value of $d$, disease control in indirect infections is not as good as in direct infections, when $d = 0.9$, such a high removal rate is still not satisfactory for the control of disease transmission. The numbers of incubation populations and infected populations were as high as 1000 and 80, respectively.

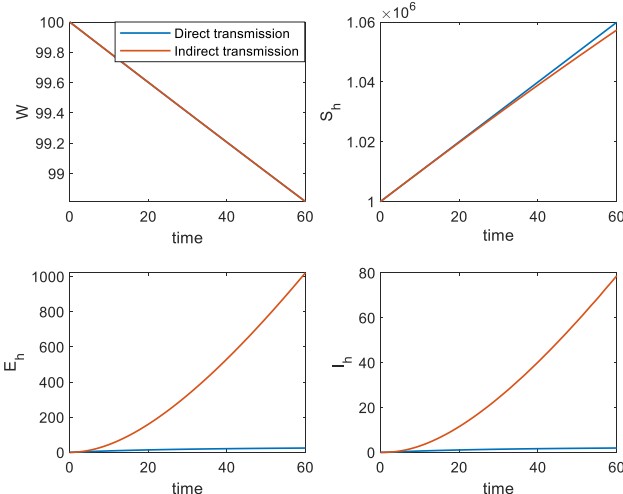

**Figure 8.** Changes in the numbers of $W$, $S_h$, $E_h$, and $I_h$ with $d = 0.9$ in model 1 (direct transmission) and 2 (indirect transmission).

The parameter D represents the removal rate of the inanimate-mediated process of the virus. This is mainly expressed as disinfection rate. In the above discussion, the disinfection rate $d_1 = 0$, and thus it is not considered. The addition of an inanimate-mediated process makes the infection more subtle, and the same changing parameters make the disease spread faster.

The above two parameters can effectively control disease transmission in direct transmission, but their effect on indirect transmission is not clear, and is insufficient for controlling disease transmission.

The following figure shows the influence of a change in $d_1$:

It is not difficult to see in Figure 9 that, by increasing $d_1$, the number of susceptible populations do not change by much, but the number of incubation population and infected population significantly decreases. Via inanimate-mediated disinfection, you can actually control the spread of the disease in the population. When the disinfection rate was increased from 0 to 0.05, the numbers of the incubation and infected populations after 60 days were controlled. When the disinfection rate is increased to 0.5, the control effect is very clear.

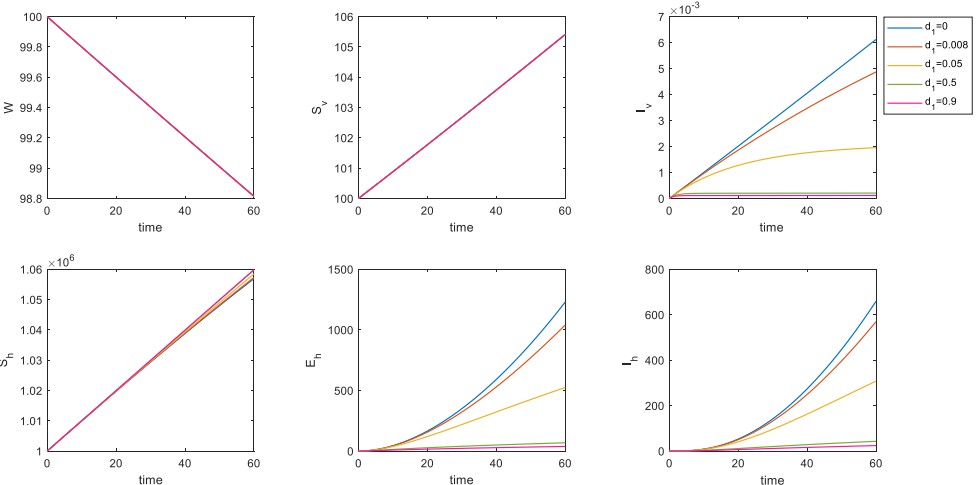

**Figure 9.** Changes in the numbers of $W$, $S_v$, $I_v$, $S_h$, $E_h$, and $I_h$ with different $d_1$ in model 2.

In order to clarify the effect of the disinfection rate, we discussed the influence of the change in disinfection rate on the number of incubation and infected populations under the values of $d_w$ and $d$ that are most conducive to disease control. When the disinfection rate is increased to 0.5, the negative effect of the inanimate-mediated can essentially be offset, that can be seen in Figures 10 and 11.

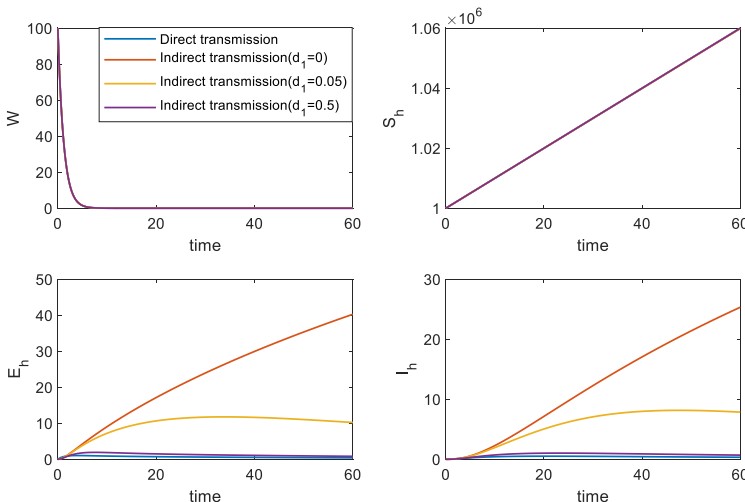

**Figure 10.** Changes in the numbers of $W$, $S_h$, $E_h$, and $I_h$ with different $d$ and $d_w = 0.85$ in model 2.

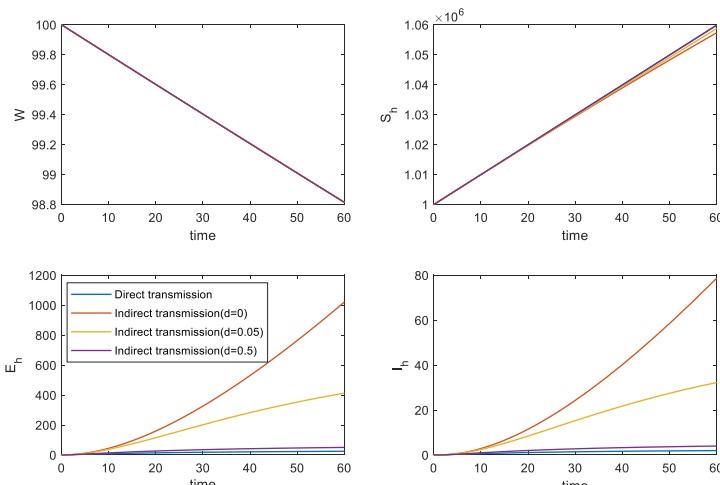

**Figure 11.** Changes in the numbers of $W$, $S_h$, $E_h$, and $I_h$ with different $d$ and $d_w = 0.9$ in model 2.

## 4. Discussion

The Occupational Safety and Health Administration declared that all AGPs pose an extremely high risk of spreading COVID-19 from patients with known or suspected infections.

The Centers for Disease Control and Prevention (CDC) reports that respiratory droplets, aerosols, and close contact (less than 1 m) are the primary modes of transmission [35]. Aerosols are small particles (less than 5–10 μm in diameter) that travel long distances, and thus are easily inhaled [8,36]. Droplets can be spread by coughing, sneezing, talking, singing, and touching mucous membranes, especially those of the nose, eyes, or mouth [46].

Contaminants in an infected person's immediate environment can carry the virus, and these contaminants are all able to spread COVID-19 [47]. Biopsies confirmed the gastrointestinal mucosal tropism of SARS-CoV-2. As the virus is present in stools, it can also be spread through the gastrointestinal tract [3,48]. This means that more objects could be infected with the virus and become vectors of transmission.

All of these conclusions indicate that, in the process of virus prevention and control, in addition to cutting off close contact between people, more attention should be paid to the elimination of the close-contact items of infected people. For high-risk industries that produce aerosols, such as dental care workers, their associated materials can easily contribute to the spread of disease.

However, in order to further clarify associated risks, more research is needed to confirm the route of transmission of SARS-CoV-2 and to measure the risk of COVID-19 among dental health professionals (DHCPs) who perform aerosol generation procedures (agp). Additionally, the effectiveness of personal protective equipment (PPE) should be evaluated to ease the concerns of the DHCP [35].

## 5. Conclusions

By comparing the two models of direct and indirect infection, the effect of the inanimate-mediated process in the early stage of infection was illustrated, which clearly increased the speed and range of virus transmission.

Regarding direct infection, the parameters $d_w$ and $d$, can accurately control the spread of the virus by limiting the entry of pathogenic populations and effective isolation of infected population, and the spread of the virus can be effectively contained. In the process of indirect transmission, the same restrictions and effective isolation are not as effective as they should be.

In order to control the spread of the virus during indirect transmission, we increased the removal rate of the virus-carrying inanimate-mediated process via inanimate-mediated disinfection. Through a numerical analysis, it is found that increasing the disinfection rate

is a very effective method to control infection. When the disinfection rate is increased to 0.5, the transmission risk caused by the inanimate-mediated process is essentially offset. When the disinfection rate increases further, the spread of the virus will be further curbed.

Therefore, it is suggested that, while limiting the entry of the pathogenic population and effective isolation of the infected population, inanimate-mediated disinfection should be carried out as necessary.

**Author Contributions:** Conceptualization, X.G. and Q.P.; methodology, X.G., S.C. and M.H.; software, X.G.; validation, L.Q., S.C. and Q.P.; formal analysis, X.G., S.C. and L.Q.; Funding acquisition, S.C.; writing—original draft preparation, X.G.; writing—review and editing, X.G. and M.H.; visualization, X.G. and L.Q.; supervision, M.H. All authors have read and agreed to the published version of the manuscript.

**Funding:** The High-level Talents Innovation Support Program of Dalian No.2021RQ062.

**Institutional Review Board Statement:** Not applicable.

**Informed Consent Statement:** Informed consent was obtained from all the subjects involved in the study.

**Data Availability Statement:** Not applicable.

**Conflicts of Interest:** The authors declare no conflict of interest.

## Appendix A

$$S_h^1 = \frac{(d_h + d)(\alpha + d_h)}{\beta_h(\alpha + d + d_h)}$$

$$E_h^1 = \frac{\beta_h A(\alpha + d + d_h) - d_h(\alpha + d_h)(d_h + d)}{\beta_h(\alpha + d_h)(\alpha + d + d_h)}$$

$$I_h^1 = \frac{\alpha(\beta_h A(\alpha + d + d_h) - d_h(\alpha + d_h)(d_h + d))}{\beta_h(\alpha + d + d_h)(\alpha + d_h)(d + d_h)}$$

$$E_h^* = \frac{A - S_h^* d_h}{\alpha + d_h}$$

$$I_h^* = \frac{\alpha(-\beta_v^2 N S_h^* d_h - \beta_v N S_h^* \beta b_v + \beta_v N S_h^* \beta d_v + \beta_v^2 AN - \beta_v S_h^* d_h b_v + d_1 S_h^* \beta b_v + d_v S_h^* \beta b_v + \beta_v A b_v)}{\beta_h \beta_v S_h^*(\alpha + d + d_h)(\beta_v N + b_v)}$$

where $S_h^*$ is the positive real root of equation.

$$\beta_h \beta_v d_h(\alpha + d + d_h)(\beta_v N + b_v)S_h^{*2}$$
$$+(-\beta_h \beta_v^2 AN\alpha - \beta_h \beta_v^2 ANd - \beta_h \beta_v^2 ANd_h - \beta_v^2 N\alpha d d_h - \beta_v^2 N\alpha d_h^2 - \beta_v^2 Nd_h^3 - \beta_v N\alpha\beta b_v d$$
$$-\beta_v N\alpha\beta b_v d_h + \beta_v N\alpha\beta d_v d + \beta_v N\alpha\beta d_v d_h - \beta_v N\beta b_v d d_h - \beta_v N\beta b_v d_h^2 + \beta_v N\beta d_v d d_h$$
$$+\beta_v N\beta d_v d_h^2 - \beta_v \beta_h A\alpha b_v - \beta_v \beta_h Adb_v - \beta_v \beta_h Ad_h b_v - \beta_v \alpha d d_h b_v - \beta_v \alpha d_h^2 b_v$$
$$-\beta_v db_v d_h^2 - \beta_v b_v d_h^3 + \alpha\beta d_1 db_v + \alpha\beta d_1 d_h b_v + \alpha\beta d_v db_v + \alpha\beta d_h d_v b_v + \beta d_1 dd_h b_v$$
$$+\beta d_1 d_h^2 b_v + \beta d_h dd_v b_v + \beta d_v d_h^2 b_v)S_h^*$$
$$+\beta_v A(d + d_h)(\alpha + d_h)(\beta_v N + b_v) = 0$$

$$S_v^{**} = -\frac{\beta_v TN + \beta_w KNb_w - \beta_w KNd_w + b_v T - b_v Nb_w + d_v Nb_w}{b_v b_w}$$

$$I_v^{**} = \frac{T}{b_w}$$

$$S_h^{**} = Q$$

$$E_h^{**} = \frac{A - Qd_h}{\alpha + d_h}$$

$$I_h^{**} = \frac{\alpha(Ab_w - QT\beta - Qd_hb_w)}{b_w\beta_h Q(\alpha + d + d_h)}$$

where $Q$ is the root of equation.

$$(\beta_h\alpha b_w d_h + \beta_h b_w d_h d + \beta_h b_w d_h^2)Q^2$$
$$+(-A\alpha\beta_h b_w - \beta_h Adb_w - \beta_h Ad_h b_w - T\alpha\beta d - T\alpha\beta d_h - T\beta dd_h - T\beta d_h^2 - \alpha d_h db_w - \alpha d_h^2 b_w - d_h^2 db_w - d_h^3 b_w)Q$$
$$+A\alpha db_w + A\alpha d_h b_w + Add_h b_w + Ad_h^2 b_w = 0$$

where $T$ is the root of equation.

$$(\beta_v^2 N + \beta_v b_v)T^2 + (2\beta_v\beta_w KNb_w - 2\beta_v\beta_w KNd_w - \beta_v Nb_v b_w + \beta_v Nd_v b_w + \beta_w Kb_v b_w - \beta_w Kb_v d_w + d_1 b_v b_w + d_v b_v b_w)T$$
$$+\beta_w^2 K^2 Nb_w^2 - 2\beta_w^2 K^2 Nb_w d_w + \beta_w^2 K^2 Nd_w^2 - \beta_w KNb_v b_w^2 + \beta_w KNb_v b_w d_w + \beta_w KNd_v b_w^2 - \beta_w KNd_v b_w d_w = 0$$

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
