# Peer review of "Influence and Control of SARS-CoV-2 Transmission under Two Different Models"

_applsci, doi:10.3390/app122211849_

Round 1

Reviewer 1 Report

Comments on the paper of Xubin Gao et.al “Influence and control of inanimate-mediated on SARS-CoV-2 transmission ”, Manuscript Number: 1987679.

 Comments and Suggestions for Authors:

The authors describe the not yet fully understood SARS-Cov-2 transmission system. They present two models of differential equations:

- the first model only discusses the virus transmission to humans directly,

- the second model introduces mediators.

The study appears to be carefully conducted with a sound analysis. The results as well as their interpretation look well justified, but the paper is not clearly written. I therefore recommend publication, however, after major changes by the authors:

a)      There is a very lack of numbers in the text of the graphs in question Fig 1, 2, 4, 5, 6.

b)     Reproduction Type:

1.

1. 1.

1. 1. 1.

is very tedious to read. Maybe it’s worth a combination.

c)    It can standardize the size of figs ..

d)    Not very extensive literature review.

 As for the minor shortcomings:

e)      Page 2, Line 76: “comma”

f)        Page 3, line 88: “Sh and Eh – too high”

g)      Page 4, line 88: “Sv and Iv – too high”

h)       Page 8, Line 200: Fig.3. –not bold

i)          Page 8, Line 210: the line - strange font

j)          Page 12, line 269: “Sv, Iv … – too high”

k)        Page 12, Line 270: Fig.9. –not bold

l)         Page 13, Line 287: 4. Conclusion – next page

Reviewer 2 Report

The present manuscript entitled “Influence and control of inanimate-mediated on SARS-CoV-2 2 transmission” aimed on the transmission of SARS-CoV-2 2 as all of us know due to this COVID-19 disease whole world suffered continuously from December 2019. Overall well written manuscript and also found scientific significance for the future research in the field of COVID-19. Authors should descried the usefulness and impact of the work. There are some grammatical mistakes needs to be corrected by authors.

Author Response

The usefulness and impact of the work have been descried. 

We have used english editing services to corrected grammatical mistakes.

Reviewer 3 Report

Dear Authors,

the paper in general is badly prepared. It requires serious corrections.

1. Starting with the introduction part, there are numbers in between the words, that means nothing to the reader - please, correct that

2. The article should begin with the information how and when did the virus occur and how did it spread. It should contain general, short information on the virus characteristics and spread. It should also contain the most frequent symptoms of the disease

- Martynowicz H, Jodkowska A, PorÄ™ba R, Mazur G, WiÄ™ckiewicz M. Demographic, clinical, laboratory, and genetic risk factors associated with COVID-19 severity in adults: A narrative review. Dent Med Probl. 2021;58(1):115–121. doi:10.17219/dmp/131795

- Paradowska-Stolarz AM. Oral manifestations of COVID-19 infection: Brief review. Dent Med Probl. 2021;58(1):123–126. doi:10.17219/dmp/131989

3. The results do not correspond to the title of the article - either you should change the title or modify results so that it is a "one piece"

4. The article should have additional parts, typical for this kind of article. Several parts are missing:

- discussion

- conclussions

- limitations

5. The discussion should contain the spread of Covid-19 with, eg. areosol generating procedures:

Manzar S, Kazmi F, Bin Shahzad H, Qureshi FA, Shahbaz M, Rashid S. Estimation of the risk of COVID-19 transmission through aerosol-generating procedures. Dent Med Probl. 2022;59(3):351–356. doi:10.17219/dmp/149342

DuÅ›-Ilnicka I, Krala E, CholewiÅ„ska P, Radwan-Oczko M. The Use of Saliva as a Biosample in the Light of COVID-19. Diagnostics (Basel). 2021 Sep 26;11(10):1769. doi: 10.3390/diagnostics11101769. 

Torul D, Omezli MM. Is saliva a reliable biofluid for the detection of COVID-19? Dent Med Probl. 2021;58(2):229–235. doi:10.17219/dmp/132515 

To sum up, the paper has serious flaws, the narration is not conducted correctly. It needs huge changes. Best regards!

Reviewer 4 Report

Although the topic is very important, authors are late one year as a lot of new findings were published in 2021 - 2022; therefore, references are also incomplete and relevant publications, e.g.  of 2022 or 2021 are missing/

Author Response

We have reorganized the introduction and added nearly 20 recent references for 2021 and 2022.

Round 2

Reviewer 1 Report

I don’t have any more questions.

Author Response

Thank you for your comments and suggestions.

Reviewer 3 Report

Dear Authors, thank you for the corrections so that the paper meets the standards of the ones for the original papers.

Here are some failures, that I still noticed:

1. check the paragraphs - the spacing between the text is different in different paragraphs

2. Lines 129-131 should be deleted - everyone should know what the result section is about, you may start with "3.1"

3. Although the limitations are usually a separate chapter, but in this case they are rather a short note, so they may stay that way.

Reviewer 4 Report

The authors have considered the most comments and addressed issues highlighted in the review process. The manuscript has been modified to take comments into account. 

The manuscript could be accepted after minor revision. Also, it should be checked over by a native English speaker before publication. More comments are indicated in the text

Author Response

Thank you for your comments and suggestions. We have used an English editing service to fix syntax errors.